# Association between gastroesophageal reflux disease and dental caries among adults in the Azar cohort population: A cross-sectional study

Zeinab Mahboobi[1,2]*, Ataollah Jalili asl[2], Nasrin Sharififard[2], Elnaz Faramarzi[1], Younes Ghavamlaleh[3]

1 Liver and Gastrointestinal Diseases Research Center, Tabriz University of Medical Sciences, Tabriz, Iran,
2 Department of Community Oral Health, Faculty of Dentistry, Tabriz University of Medical Sciences, Tabriz, Iran, 3 Department of Emergency Medicine, Faculty of Medicine, Islamic Azad University of Tabriz, Tabriz, Iran

* znbmhb@gmail.com

## Abstract

### Background

Considering the high prevalence of dental caries in adults, it is necessary to study its risk factors to prevent the disease. Gastroesophageal reflux disease (GERD) is a common chronic disease with an increasing incidence that may affect the quality and quantity of saliva.

### Objectives

This study aimed to determine the association of gastroesophageal reflux disease (GERD) with dental caries according to the DMFT index in the Azar cohort population.

### Materials and methods

This cross-sectional study was performed on data obtained from the enrollment phase of the Azar Cohort Study (ACS), conducted on 15,000 adults aged 35 to 70. Data of 905 subjects with gastroesophageal reflux disease in the ACS—without full denture—together with a control group of 1894 subjects, who were matched in terms of gender and age, were included in this study. Dental caries data and data on GERD, demographic and socioeconomic characteristics, individual and nutritional habits, BMI, and toothbrushing frequency were elicited from the databank of the ACS center. A Generalized Linear Models assuming negative binomial distribution with a log-link function was used for analytical statistics to examine the effect of possible confounding variables.

### Results

The DMFT index in the group with GERD was 15.09±6.18, and for the control group was 15.00±6.07. No statistical association was seen between GERD and dental caries. Among

**Data Availability Statement:** All relevant data are within the paper and its Supporting Information files.

**Funding:** The author(s) received no specific funding for this work.

**Competing interests:** The authors have declared that no competing interests exist.

the variables included in the regression analysis, being younger and toothbrushing one or more times per day were associated with a lower DMFT index score.

## Conclusion

According to the results, having GERD did not increase the risk of dental caries. However, due to the cross-sectional design of the study, the results should be interpreted cautiously. The results showed that oral hygiene is one of the most influential factors in reducing dental caries prevalence.

## Introduction

Although dental caries is usually preventable, many people worldwide still suffer from its pain and discomfort, and it imposes high medical costs on individuals and societies [1]. According to previous reports, oral disorders are still a public health challenge [2], and dental caries remains one of the leading health problems in most countries, affecting most adults [3]. A study in 2007 [4] showed that the DMFT index in Iranian adults aged 34–44 was 11±6.4, which was in the range reported for this age group by the World Health Organization (WHO) in 2004. Approximately 33% of Iranian adults [4] and 35% of the world's population had untreated caries in permanent teeth, so it has remained the most common public health problem in the last three decades [5]. Differences in oral health status in different societies indicate that behavioral aspects and various socioeconomic and environmental factors prevail in countries and within populations [3].

Dental caries is a complex and multifactorial disease with several proximal and distal determinants. The main risk factors for dental caries are diet, a susceptible host, and microorganisms which act in a complex context of socioeconomic, behavioral, environmental, and demographic factors. In fact, individual behaviors such as oral hygiene habits or dietary patterns are determined by socioeconomic status in which people are born, grow up, work, and age. These conditions provide the opportunities for and limit individual behaviors [6].

It is indicated that some systemic diseases are related to dental caries development. Gastroesophageal reflux disease (GERD) is a condition that may potentially cause tooth decay due to the changes it creates in the oral cavity, such as the oral pH drop, decreased salivation, and disruption of its buffering capacity [7]. Gastroesophageal reflux is a physiological process of effortless backward movement of the contents of the stomach to the esophagus; in contrast, gastroesophageal reflux disease (GERD) occurs when the rate of gastroesophageal reflux is higher than expected and is recognized by both classic and atypical symptoms [8]. Regurgitation and heartburn are usual symptoms of GERD [9]. GERD risk factors are divided into two categories: non-modifiable risk factors, including age, sex, ethnicity, and modifiable risk factors, including lifestyle, dietary habits, and body weight [10]. Gastroesophageal reflux disease is a highly prevalent disease with an estimated prevalence of 43.07% in the Iranian population [11]. The overall approximate daily prevalence of GERD has been reported to be 10–20% in Europe and the USA and less than 5% in Asia [12], with 5.64% (95% CI: 3.77–8.35) in the Iranian population [11]. A review showed that the incidence of GERD is increasing mainly in North America and East Asia [13], and it is expected to continue to increase due to new dietary habits and lifestyles [7].

The minerals of the tooth surface are in a dynamic equilibrium with the oral fluids, and the enamel of teeth demineralizes and remineralizes several times a day [6]. Caries develop when

the balance between these phases is disturbed, and demineralization overcomes remineralization. Saliva is one of the efficient factors in neutralizing the acids produced by bacteria and helps the remineralization phase [6]. Some studies have shown that the quality and quantity of saliva [14, 15] change in patients with GERD. Studies have reported conflicting results regarding the association of GERD with caries. Some studies have reported more caries in individuals with GERD, attributing it to several factors, including changes in the composition and volume of saliva or immunological changes [14, 16, 17]. Some other studies have shown that *Streptococcus mutans* counts decrease due to a significant drop in salivary pH in patients with GERD, leading to the conclusion that this factor can reduce the incidence of caries in this group [18–20]. On the other hand, some studies have shown no association between the disease and an increase in dental caries rate [21–23]. Considering the high prevalence of dental caries and GERD and contradictions in the relationship between dental caries and GERD in previous studies, the present study aimed to determine the association of gastroesophageal reflux disease (GERD) and dental caries in the Azar cohort population. This study hypothesizes that there is no association between having GERD and dental caries development, considering potential confounders.

## Methods

### Design and setting

All the data used in the present analytical cross-sectional study were obtained from the enrollment phase of the Azar Cohort Study (ACS). The ACS was framed in three phases: (1) the pilot phase; (2) the enrolment of participants phase; and (3) three follow-up phases for 15 years. It is a large epidemiological study with a sample size of 15,000 subjects aged 35–70 in the northwest of Iran in Shabestar, a city in East Azerbaijan Province. The ACS is part of a large prospective longitudinal project called the Persian Cohort (Prospective Epidemiological Research Studies in Iran), which aims to collect data about the relevant risk factors of prevalent non-communicable diseases (NCDs) in Iran [24, 25]. Four valid and reliable questionnaires consisting of 482 items were used to collect data in the Persian cohort, and face-to-face interviews and clinical examinations were used to complete these questionnaires, and the information was registered online in the cohort study database [25, 26].

The recruitment phase of the ACS started in October 2014 and continued until 2015. The profile of the ACS and research methods have been described previously [26]. The ACS was approved by the Ethical Committee of Tabriz University of Medical Sciences (record number: tbmed.rec.1393.205). Ethical approval of the present study was received from the Ethics Committee of Tabriz University of Medical Sciences (IR.TBZMED.REC.1401.080) on April 2022, and after that, the Azar Cohort Center provided access to data needed in this study [24–26].

### Participants and sample size

In the present study, data from subjects with a history of GERD in the ACS were recruited. The GERD group consisted of patients who provided positive answers to these questions from the medical questionnaire of the ACS: "Have you ever been diagnosed with gastroesophageal reflux disease?" and "Have you had reflux of food from the stomach to the esophagus in the past year?". A total of 2010 patients with GERD were identified in the Azar cohort population. Subjects with full dentures were excluded from this study, and data of 905 subjects without full dentures qualified for the GERD group. Also, a comparison group, including subjects without GERD, who had no full dentures and were matched in terms of gender and age, was selected randomly from the ACS databank. Considering 1894 subjects in the control group, data from

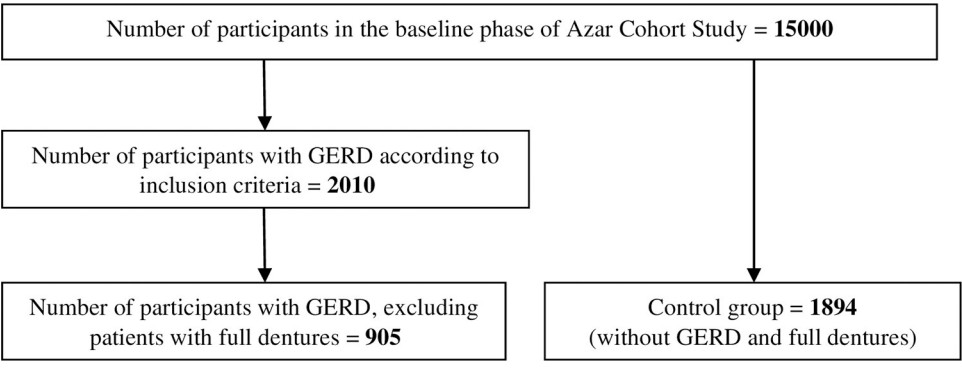

**Fig 1. Flow diagram of the number of study subjects.**

2799 participants were included in this study. Fig 1 presents the selection process of the study subjects.

## Variables and data measurement

All the data for this study were obtained from the databank of the ACS. One general, two medical, and one nutrition questionnaire were used to collect information in the Azar cohort study [25, 26]. Among the variables of the Azar cohort study used in the present study, dental caries and body mass index (BMI) were measured based on clinical examinations by trained examiners. Dental caries was recorded as the number of decayed, missing, and filled teeth (DMFT index) according to WHO criteria [27], and the examiner was uninformed of the systemic conditions of the participants. GERD history and the frequency of regurgitation in the ACS were collected using a medical questionnaire by face-to-face interview method and based on the participants' self-report. Information on demographic characteristics (age and gender) and socioeconomic conditions (educational level and wealth score index) was gathered by the general questionnaire. According to the Persian Cohort Study, to measure economic status, the wealth score index was recorded based on long-lasting belongings and housing characteristics. Information related to behavioral habits (toothbrushing frequency, tobacco use, alcohol drinking, and consumption of free sugars) was extracted from the medical questionnaire and the food frequency questionnaire [26].

In the present study, the frequency of regurgitation was divided into three groups based on the data distribution in the samples, including once or more/week, 2–3 times/month, and no regurgitation [28]. The educational level was classified into two groups: (1) illiterate, elementary school, and middle school (0–8 years of schooling) and (2) high school, high school graduate, and higher education, including associate degree, bachelor's degree, master's degree, and doctorate (≥9 years of schooling). The wealth score index was classified into five groups from the poorest to the richest (very poor ≤ -.7966159, poor = -.7966160 to -.3232089, moderate = -.3232090 to .2767860, good = .2767861 to .8069400, and very good = .8069401+). The measurement of food consumption in the food frequency questionnaire of the ACS was based on the amount and frequency of daily, weekly, monthly, and yearly consumption. Categorizing the free sugars consumption (gr/day) was based on the data distribution in the sample (Sweet foods: ≤10.34, 10.35–23.90, and >23.91; sweet drinks: ≤18.93, 18.94–62.08, and ≥62.09; natural fruit juice: ≤0.000, 0.001–7.56 and ≥7.57). The body mass index was categorized into three groups: overweight (≥25), normal (18.5–24.9), and underweight (<18.5).

## Statistical analysis

Data analysis was conducted by IBM SPSS Statistics for Windows, version 20 (IBM Corp., Armonk, N.Y., USA) at a significance level of <0.05. In the present study, the DMFT index was used as the outcome variable. Independent sample t-test and one-way ANOVA were used to compare the mean DMFT scores between subgroups.

Considering the DMFT index as a count data and according to its distribution, the negative binomial with log-link function in Generalized Linear Models (GLM) analysis was used to analyze the relationship between dental caries and the independent variables. According to the aim of this study, the main independent variables that might be affecting dental health were GERD and the frequency of food regurgitation. The variables potentially affecting this relationship, considering confounding effects, include demographic factors (age and gender), socioeconomic status (wealth score index and educational level), behavioral habits (tobacco use, alcohol consumption, frequency of toothbrushing, consumption of sweet foods and beverages containing free sugars), and BMI. The variables whose p-value was <0.2 in the univariate analysis were considered for being included in the multiple regression analysis. The variable selection process for the multiple regression was based on the backward-stepwise method.

# Results

## Description of the sample

In the study population of the Azar cohort, 905 subjects with GERD did not have full dentures and entered as GERD group. In the present study, more than 60% of the participants were women. About half were in the age range of 35–45 years. Half of the participants had less than 9 years of education, and this was 63% in people with GERD. Most participants belonged to the middle or higher groups regarding the wealth score index. Considering individual habits, around 12% of subjects used tobacco, 2% used to have alcoholic drinks, and 60% brushed their teeth once or more per day. In the group with GERD, about half of the participants brushed their teeth once or more per day. Two-thirds of participants did not report regurgitation symptoms, and nearly 80% were overweight or obese. The DMFT index score varied between 0 and 32 for the participants. The mean DMFT (SD) in the groups with GERD and the control group were 15.09 (6.18) and 15.00 (6.07), respectively. Table 1 shows the frequency distributions of the evaluated variables in the subjects with GERD and control groups.

## Analytical results

Although the DMFT index was higher in the group suffering from GERD than in the control group, both compare mean analysis and univariate regression analysis indicated no statistical association between GERD and dental caries. Table 2 presents the mean ± SD for DMFT in different subgroups of independent variables separately for the group suffering from GERD and the control group.

In the univariate analysis, a significant relationship was observed between the DMFT index and the variables of educational level (p = 0.02) and frequency of toothbrushing (p≤0.001). The variables whose p-value was < 0.2 and included in the multiple regression analysis were age at the interview, years of education, tobacco use, and toothbrushing frequency.

The results of the multiple regression analysis showed a significant association between the outcome variable and toothbrushing frequency so that lack of daily toothbrushing was associated with an increase in dental caries prevalence of approximately 4-fold [IRR = 4.43 (95% CI: 2.85–6.90), p≤0.001]. Also, in this study, there was significantly less dental caries in younger age groups of 35–45 years and 46–55 years [IRR = 0.01 (95% CI: 0.00–0.09) p≤0.001 &

**Table 1. Characteristics of the participants in the GERD and control groups.**

| Variables | Categories | GERD group (N %) N = 905 | Control group (N %) N = 1894 | Total (N %) N = 2799 |
|---|---|---|---|---|
| *Gender* | | | | |
| | Female | 560 (61.88) | 1158 (61.14) | 1718 (61.38) |
| | Male | 345 (38.12) | 736 (38.86) | 1081 (38.62) |
| *Age at interview* | | | | |
| | 35–45 | 482 (53.26) | 992 (52.37) | 1474 (52.66) |
| | 46–55 | 323(35.69) | 662 (34.95) | 985 (35.19) |
| | 56–65 | 90 (9.94) | 214 (11.30) | 304 (10.86) |
| | ≥66 | 10 (1.11) | 26 (1.38) | 36 (1.29) |
| *Wealth score index* | | | | |
| | Very poor | 146 (16.13) | 230 (12.14) | 376 (13.42) |
| | Poor | 117 (12.93) | 178 (9.40) | 295 (10.54) |
| | Moderate | 201 (22.21) | 343 (18.11) | 544 (19.44) |
| | Good | 224 (24.15) | 491 (25.92) | 715 (25.54) |
| | Very good | 217 (23.98) | 652 (34.42) | 869 (31.05) |
| *Years of education* | | | | |
| | 0–8 | 569 (62.87) | 852 (44.98) | 1421 (50.77) |
| | ≥9 | 336 (37.12) | 1042 (55.02) | 1378 (49.23) |
| *Tobacco use* | | | | |
| | Yes | 106 (11.71) | 234 (12.35) | 340 (12.15) |
| | No | 799 (88.29) | 1660 (87.65) | 2459 (87.85) |
| *Alcohol drinking* | | | | |
| | Yes | 17 (1.88) | 40 (2.11) | 57 (2.04) |
| | No | 888 (98.12) | 1854 (99.89) | 2742 (97.96) |
| *Toothbrushing* | | | | |
| | <1/day | 455 (50.28) | 668 (35.27) | 1123 (40.12) |
| | ≥1/day | 450 (49.72) | 1226 (64.73) | 1676 (59.88) |
| *Sweet foods (gr/day)* | | | | |
| (Free sugars: gr/day) | >23.91 | 268 (29.61) | 654 (34.55) | 922 (32.95) |
| | 10.35–23.90 | 320 (35.36) | 601 (31.75) | 921 (32.92) |
| | ≤10.34 | 317 (35.03) | 638 (33.70) | 955 (34.13) |
| *Sweet beverages* | | | | |
| (Free sugars: gr/day) | ≥62.09 | 243 (26.85) | 644 (34.02) | 887 (31.70) |
| | 18.94–62.08 | 280 (30.94) | 606 (32.01) | 886 (31.67) |
| | ≤18.93 | 382 (42.21) | 643 (33.97) | 1025 (36.63) |
| *Natural fruit juice* | | | | |
| (Free sugars: gr/day) | ≥7.563 | 260 (28.73) | 638 (33.70) | 898 (32.09) |
| | .001–7.562 | 235 (25.97) | 442 (23.35) | 677 (24.20) |
| | ≤0.000 | 410 (45.30) | 813 (42.95) | 1223 (43.71) |
| *Body Mass Index (kg/m$^2$)* | | | | |
| | <18.5 | 2 (0.22) | 11 (0.58) | 13 (0.46) |
| | 18.5–24.9 | 167 (18.45) | 425 (22.44) | 592 (21.15) |
| | ≥ 25 | 736 (81.33) | 1458 (76.98) | 2194 (78.39) |
| *Frequency of regurgitation* | | | | |
| | Once or more/week | 60 (6.63) | 45 (2.38) | 105 (3.75) |
| | 2–3 times/month | 154 (17.02) | 249 (13.14) | 403 (14.40) |
| | No | 691 (76.35) | 1600 (84.48) | 2291 (81.85) |
| *DMFT (Mean ± SD)* | | 15.09 ± 6.18 | 15.00 ± 6.07 | 15.03 ± 6.10 |

**Table 2. The mean ± SD of DMFT in independent variables in the GERD and control groups.**

| Variables | Categories | GERD group Mean ± SD | p-value | Control group Mean ± SD | p-value |
|---|---|---|---|---|---|
| *Gastroesophageal reflux disease* | | | | | 0.73[€] |
| | Yes | 15.09 ± 6.18 | | ------ | |
| | No | ------ | | 15.00 ± 6.07 | |
| *Frequency of regurgitation* | | | 0.87* | | |
| | Once or more/week | 14.74 ± 5.32 | | 14.70 ± 5.33 | 0.57* |
| | 2–3 times/month | 14.91 ± 5.80 | | 15.36 ± 6.08 | |
| | No | 15.15 ± 6.31 | | 14.94 ± 6.07 | |
| *Gender* | | | | | |
| | Female | 14.54 ± 5.60 | ≤0.001[€] | 14.91 ± 5.60 | 0.42[€] |
| | Male | 15.97 ± 6.93 | | 15.15 ± 6.74 | |
| *Age at interview* | | | | | |
| | 35–45 | 13.51 ± 5.73 | ≤0.001* | 13.75 ± 5.57 | ≤0.001* |
| | 46–55 | 16.15 ± 5.75 | | 15.87 ± 6.13 | |
| | 56–65 | 19.28 ± 6.72 | | 17.73 ± 6.46 | |
| | ≥66 | 18.90 ± 9.40 | | 18.04 ± 7.27 | |
| *Wealth score index* | | | | | |
| | Very poor | 15.84 ± 6.62 | 0.02* | 16.05 ± 6.51 | 0.08* |
| | Poor | 16.19 ± 6.06 | | 15.17 ± 6.44 | |
| | Moderate | 15.15 ± 6.12 | | 14.87 ± 6.15 | |
| | Good | 14.13 ± 6.24 | | 14.74 ± 5.99 | |
| | Very good | 14.91 ± 5.81 | | 14.85 ± 5.78 | |
| *Years of education* | | | | | |
| | 0–8 | 15.72 ± 6.33 | ≤0.001[€] | 15.70 ± 6.34 | ≤0.001[€] |
| | ≥9 | 14.01 ± 5.77 | | 14.43 ± 5.77 | |
| *Tobacco use* | | | | | |
| | Yes | 17.19 ± 7.04 | ≤0.001[€] | 16.29 ± 7.27 | ≤0.001[€] |
| | No | 14.81 ± 6.00 | | 14.82 ± 5.84 | |
| *Alcohol drinking* | | | | | |
| | Yes | 15.12 ± 7.92 | 0.22[€] | 16.55 ± 7.91 | 0.98[€] |
| | No | 15.09 ± 6.15 | | 14.97 ± 6.02 | |
| *Toothbrushing* | | | | | |
| | <1/day | 15.96 ± 6.61 | ≤0.001[€] | 16.03 ± 6.75 | ≤0.001[€] |
| | ≥1/day | 14.21 ± 5.58 | | 14.44 ± 5.58 | |
| *Sweet foods (gr/day)* *(Free sugars: gr/day)* | >23.91 | 14.68 ± 6.09 | 0.08* | 15.12 ± 5.94 | 0.82* |
| | 10.35–23.90 | 14.81 ± 5.91 | | 14.95 ± 6.09 | |
| | ≤10.34 | 15.71 ± 6.48 | | 14.93 ± 6.18 | |
| *Sweet beverages* *(Free sugars: gr/day)* | ≥62.09 | 15.18±6.15 | 0.69* | 15.24 ± 6.16 | 0.46* |
| | 18.94–62.08 | 14.83±5.93 | | 14.89 ± 5.95 | |
| | ≤18.93 | 15.22±6.38 | | 14.87 ± 6.12 | |
| *Natural fruit juice* *(Free sugars: gr/day)* | ≥7.563 | 14.91 ± 6.49 | 0.69* | 14.80 ± 6.13 | 0.56* |
| | 0.001–7.562 | 14.94 ± 6.38 | | 15.15 ± 6.52 | |
| | ≤0.000 | 15.28 ± 5.86 | | 15.09 ± 5.75 | |
| *Body Mass Index (kg/m²)* | | | | | |

*(Continued)*

**Table 2.** (Continued)

| Variables | Categories | GERD group Mean ± SD | p-value | Control group Mean ± SD | p-value |
|---|---|---|---|---|---|
| | <18.5 | 18.00 ± 2.83 | 0.33* | 16.55 ± 6.99 | 0.69* |
| | 18.5–24.9 | 15.65 ± 6.34 | | 14.94 ± 6.23 | |
| | ≥25 | 14.95 ± 6.14 | | 15.01 ± 6.01 | |

*One-way ANOVA

€ Independent sample t-test

IRR = 0.12 (95% CI: 0.02–0.82) p = 0.03, respectively]. Table 3 presents the results of univariate and multiple regression analysis using negative binomial regression to determine the effect of independent variables on the DMFT index.

## Discussion

The most common dental complication reported in patients with gastroesophageal reflux disease (GERD) is tooth erosion [29, 30]; however, few studies have investigated the relationship between this medical condition and dental caries [19–23]. The present study aimed to determine the association between GERD and dental caries based on the DMFT index, considering potential confounding variables through regression analysis. There was no significant association between GERD and a higher rate of dental caries compared with healthy subjects. The results showed no significant association between DMFT index score and the frequency of regurgitation, gender, socioeconomic status, BMI, use of tobacco and alcohol, and consumption of sweet foods, sweet drinks, and natural fruit juice. However, a significant relationship was observed between lower DMFT and being younger and daily toothbrushing frequency.

Our study did not show any significant relationship between GERD and a higher rate of dental caries, which was in line with the results of studies by Watanabe, Sîmpălean, and Munoz [21–23]. However, some studies revealed a possibility of increased dental caries in patients with GERD due to decreased salivary flow, swallowing dysfunction and low salivary buffering capacity [14, 16]. It has been reported that some other factors like parafunctional habits, such as bruxism, or the individual's lifestyle, such as improper oral hygiene, play a role in increasing caries in patients with GERD [31, 32]. A study by Borysenko et al. indicated that the prevalence and intensity of dental caries were high in patients with GERD compared with healthy controls and concluded that it could be attributed to immunological changes in GERD patients [17]. On the other hand, some studies showed lower dental caries rates in patients with GERD compared with healthy subjects as a result of dramatic oral pH drops due to acid reflux. These studies indicated that pH drop leads to a reduction in the population of the bacteria, including *Streptococcus mutans* colonies, as the main microbial aetiological factor in the development of tooth decay. Filipi explained that although *S. mutans* can survive in pH values <4.2, the pH decline in the oral cavity of GERD might be so remarkable that it can stop the metabolic activity of *S. mutans* [18–20]. It might be concluded that in patients suffering from GERD, despite the decrease in the quantity and quality, and buffering capacity of saliva, which can lead to a higher risk of caries, the decline in the population of cariogenic bacteria in the oral cavity leads to a lower risk of caries. The combination of these two factors possibly leads to the absence of a significant relationship between GERD and dental caries.

The present study showed a significant relationship between the participants' age and the DMFT index score, with less dental caries in the younger age groups, consistent with other studies. This finding might be attributed to the fact that the effects of risk factors in caries

**Table 3. Univariate and multiple regression analysis of the association between DMFT index and independent variables using the negative binomial with log-link function in Generalized Linear Models (n = 2799).**

| Variables | Categories | IRR* | p-value | IRR* | p-value |
|---|---|---|---|---|---|
| *Gastroesophageal reflux disease* | | | | | |
| | Yes | 1.01 (0.93–1.09) | 0.89 | | |
| | No | 1 | | | |
| *Frequency of regurgitation* | | | | | |
| | Once or more/week | 1.04 (0.85–1.28) | 0.68 | | |
| | 2–3 times/month | 1.00 (0.90–1.12) | 0.98 | | |
| | No | 1 | | | |
| *Gender* | | | | | |
| | Female | 0.96 (0.89–1.04) | 0.30 | | |
| | Male | 1 | | | |
| *Age at interview*€ | | | | | |
| | 35–45 | 0.74 (0.53–1.05) | 0.09 | 0.01 (0.00–0.09) | ≤0.001 |
| | 46–55 | 0.87 (0.62–1.23) | 0.44 | 0.12 (0.02–0.82) | 0.03 |
| | 56–65 | 0.99 (0.70–1.42) | 0.98 | 1.06 (0.14–7.97) | 0.96 |
| | ≥66 | 1 | | 1 | |
| *Wealth score index* | | | | | |
| | Very poor | 1.07 (0.95–1.22) | 0.26 | | |
| | Poor | 1.05 (0.91–1.20) | 0.51 | | |
| | Moderate | 1.01 (0.90–1.13) | 0.90 | | |
| | Good | 0.98 (0.88–1.08) | 0.68 | | |
| | Very good | 1 | | | |
| *Years of education*€ | | | | | |
| | 0–8 | 1.10 (1.02–1.18) | 0.02 | | |
| | ≥9 | 1 | | | |
| *Tobacco use*€ | | | | | |
| | Yes | 1.12 (0.99–1.26) | 0.06 | | |
| | No | 1 | | | |
| *Alcohol drinking* | | | | | |
| | Yes | 1.07 (0.82–1.41) | 0.60 | | |
| | No | 1 | | | |
| *Toothbrushing*€ | | | | | |
| | <1/day | 5.04 (3.19–7.97) | ≤0.001 | 4.43 (2.85–6.90) | ≤0.001 |
| | ≥1/day | 1 | | 1 | |
| *Sweet foods* | | | | | |
| *(Free sugars: gr/day)* | >23.91 | 0.82 (0.47–1.43) | 0.49 | | |
| | 10.35–23.90 | 0.75 (0.43–1.31) | 0.31 | | |
| | ≤10.34 | 1 | | | |
| *Sweet beverages* | | | | | |
| *(Free sugars: gr/day)* | ≥62.09 | 1.26 (0.73–2.17) | 0.41 | | |
| | 18.94–62.08 | 0.88 (0.51–1.53) | 0.65 | | |
| | ≤18.93 | 1 | | | |
| *Natural fruit juice* | | | | | |
| *(Free sugars: gr/day)* | ≥7.563 | 0.73 (0.43–1.23) | 0.23 | | |
| | 0.001–7.562 | 0.93 (0.52–1.65) | 0.80 | | |
| | ≤0.000 | 1 | | | |
| *Body Mass Index (kg/m²)* | | | | | |

*(Continued)*

**Table 3.** (Continued)

| Variables | Categories | IRR* | p-value | IRR* | p-value |
|---|---|---|---|---|---|
| | ≥25 | 0.99 (0.90–1.09) | 0.83 | | |
| | <18.5 | 1.11 (0.63–1.95) | 0.72 | | |
| | 18.5–24.9 | 1 | | | |

*Incidence Rate Ratio

€ Entered in the multiple regression analysis

development are cumulative and increase with age [6, 23, 33, 34]. In addition to age, a statistically significant relationship was observed between brushing teeth once a day or more and lower DMFT index scores. Toothbrushing is a well-established method to regularly clean the teeth from microbial plaque, as one of the main aetiological factors of caries development [35].

Lukacs believed that gender differences affect dental caries prevalence through both biological (genetics, hormones, and reproductive history) and anthropological (behavioral) factors, with women exhibiting more dental caries [36]; however, in the present study, no relationship was detected between gender and dental caries, consistent with a study by Abbass [37]. Also, there was no significant relationship between the DMFT index score and socioeconomic status. A systematic review by Costa et al. in 2012 indicated that the educational level, occupation, and subject's income are related to tooth decay [38]. On the other hand, a systematic review by Reisine and Psoter in 2001 showed a weaker association between socioeconomic conditions and dental caries in adults due to fewer studies and methodological limits [39]. Schwendicke reported a stronger relationship between socioeconomic status and dental caries in developed countries [40]. Developed and developing countries were defined according to the 2008 World Bank classification. It is commonly used to categorize the world into "low and middle-income" (developing) and "high-income" countries (developed) [41]. Therefore, one justification for the non-significant relationship in our study may be that Iran is a developing country. Also, due to the nature of the study, in which data were collected with questionnaires and based on the self-report, the answers to the questions related to socioeconomic conditions may not be completely reliable.

The present study showed no association between BMI and DMFT scores. A systematic review by Silva et al. in 2013 showed insufficient evidence to prove the relationship between obesity and caries [42]. Most systematic studies concluded that there is no consensus regarding the relationship between BMI and dental caries and suggested the need for properly designed studies in this field [43–45].

Our study showed no significant relationship between the behavioral risk factors mentioned in this study (sweet foods, sweet drinks, natural fruit juices, and tobacco and alcohol use) and DMFT scores. Few studies that measure the relationship between intake of sugary substances and dental caries have been conducted in the adult age group [46]. Although most of these studies showed a significant relationship between sugar consumption and tooth decay [46, 47], some reported no association [48–50], consistent with the present study. This heterogeneity in the results of the studies may be due to confounding factors, such as concurrent consumption of caries-protective foods at meals or differences in fluoride exposure [6]. A Study by Voelker showed that smoking can affect dental health through its influence on the buffering capacity of saliva and levels of secretory IgA [51]; however, the present study showed no significant association between DMFT and smoking. Despite the positive relationship between tobacco use and dental caries in some systematic reviews, they mentioned that this relationship might not be valid due to the poor quality of the studies [52, 53]. Few studies have

examined the relationship between alcohol consumption and tooth decay. In contrast with our results, some studies showed a significant increase in the DMFT index and alcohol use [54–56]. On the other hand, according to some studies, the high fluoride content in alcoholic beverages is a barrier to new caries development [57, 58].

This research was based on a population-based study with a large sample size and can be representative of the society. However, our study had some limitations. Because this study is cross-sectional, it only shows the relationship/lack of the relationship between desired variables, and the findings are not sufficient to establish a cause-and-effect association. Information about GERD was collected through a questionnaire and based on the participants' self-report. Also, the duration of gastroesophageal reflux disease was not investigated in the ACS; therefore, the results should be interpreted cautiously.

In conclusion, our study did not show a significant association between GERD and dental caries based on the DMFT index score. As discussed, the results of other studies in this area were not consistent. The high variability between the results of different studies can be related to several factors, including the frequency of regurgitation, the duration of GERD, swallowing disorders, the buffering capacity of saliva, and the design of studies. Prospective longitudinal studies with a sufficient sample size are necessary to reach a better conclusion. There is serious concern about dental caries in Iran, mainly due to the lack of coherent preventive programs. Emphasis on oral and dental hygiene, including regular toothbrushing once a day or more, can be among the most influential factors in reducing dental caries.

## Supporting information

**S1 Dataset.**
(XLS)

## Acknowledgments

The authors are grateful for the support of Tabriz University of Medical Sciences and the Liver and Gastrointestinal Diseases Research Center. The authors also are deeply indebted to all the subjects participated in this study. We appreciate the contribution of the investigators and the staff of the Azar cohort study. We thank the close collaboration of the Shabestar health center. In addition, we would like to thank the Persian cohort study staff for their technical support.

## Author Contributions

**Conceptualization:** Zeinab Mahboobi, Younes Ghavamlaleh.

**Data curation:** Elnaz Faramarzi.

**Formal analysis:** Zeinab Mahboobi, Nasrin Sharififard, Elnaz Faramarzi.

**Methodology:** Zeinab Mahboobi, Ataollah Jalili asl, Nasrin Sharififard, Elnaz Faramarzi, Younes Ghavamlaleh.

**Supervision:** Zeinab Mahboobi.

**Writing – original draft:** Zeinab Mahboobi, Ataollah Jalili asl, Nasrin Sharififard, Elnaz Faramarzi, Younes Ghavamlaleh.

**Writing – review & editing:** Zeinab Mahboobi, Ataollah Jalili asl, Nasrin Sharififard, Elnaz Faramarzi, Younes Ghavamlaleh.

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
