## [Decision Letter · Decision Letter 0]

21 May 2023

PONE-D-23-10589Association between gastroesophageal reflux disease and dental caries among adults in the Azar cohort population: A cross-sectional studyPLOS ONE

Dear Dr. Mahboobi,

Thank you for submitting your manuscript to PLOS ONE. After careful consideration, we feel that it has merit but does not fully meet PLOS ONE’s publication criteria as it currently stands. Therefore, we invite you to submit a revised version of the manuscript that addresses the points raised during the review process.

We look forward to receiving your revised manuscript.

Kind regards,

Hadi Ghasemi

Academic Editor

PLOS ONE

Journal Requirements:

“The author(s) received no specific funding for this work.

Data used in the present study was obtained from the database of the Azar Cohort study. The Azar cohort study was supported by the liver and gastrointestinal diseases research center (Grant number 700/108 on 14 March 2016) at Tabriz University of Medical Sciences. The Iranian Ministry of Health and Medical Education has contributed to the funding used in the PERSIAN cohort through Grant no.700/534. The funders had no role in the study design, data analysis, interpretation, and writing the manuscript in this study.“

Reviewers' comments:

Reviewer's Responses to Questions

**Comments to the Author**

1. Is the manuscript technically sound, and do the data support the conclusions?

Reviewer #1: No

Reviewer #2: Partly

2. Has the statistical analysis been performed appropriately and rigorously? 

Reviewer #1: No

Reviewer #2: Yes

3. Have the authors made all data underlying the findings in their manuscript fully available?

Reviewer #1: No

Reviewer #2: No

4. Is the manuscript presented in an intelligible fashion and written in standard English?

Reviewer #1: No

Reviewer #2: Yes

5. Review Comments to the Author

Reviewer #1: Thank you for asking me to review the manuscript titled “Association between gastroesophageal reflux disease and dental caries among adults in the Azar cohort population. A cross-sectional study”

The article is relevant to the field of Internal medicine and Dentistry. However, there are flaws in the write -up;

Introduction

A major drawback of this manuscript is that the authors failed to review literature on GERD and dental caries. In addition, they didn’t identify any gap/s in literature that their study would want to address.

Generally, the paragraphs don’t flow into each other – the previous paragraph doesn’t flow to the next.

Methods

In addition to other confounding variables, the authors should have considered the oral hygiene status of the participants.

Similarly, there is no information on the fluoride exposure of the participants in the study.

Although information on free sugar consumption was obtained, the frequency of free sugar consumption between meals will be more appropriate in relation to caries formation.

Classification of GERD frequency of regurgitation should be referenced

Results

It would have been better to compare the mean DMFT scores of both groups based on the different variables (socio-demographic, oral health practices and BMI) before the regression analysis.

Discussion

This is inadequate- the authors didn’t discuss the reasons behind their findings

Reviewer #2: Title: It is limited, as there are many other factors that have been checked other than GERD. So you need to modify your title accordingly.

Abstract:

- Objectives: There are many other objectives that were not mentioned.

- Materials and methods: very confusing, as I don’t understand if all your data obtained from dataset

previously recorded, Or the clinical examination and the interviews were conducted as part of this

study. This same comment is applied to methodology section that needs to be re-written.

Introduction:

Line 53: move word ‘still’ after ‘are’.

Line 60: consistency in writing word ‘socioeconomic’.

Next to first paragraph, additional paragraph should be added to briefly preset the socioeconomic and environmental factors that is related or may affect dental caries.

Line 78: Add reference.

Add null hypothesis to the last paragraph

Methods:

-Very confusing as mentioned previously, which data that have been obtained from previous records and which data were obtained by clinical examination and questionnaires in this study??

-And if questionnaires were used, you have to specify is it online , face to face interview, paper based?

-What was the response rate in each questionnaire, number of items in each questionnaire, how these items were validated, any piloting for the questionnaire was done?

-Why 905 were selected from 2010 patients with GERD, and how these patients were selected.

-GERD patients only were diagnosed based on the two questions that have been mentioned.

- What do you mean by oral examiner, this is not acceptable description.

- Variables and data measurement section is not clear and more details should be added.

- Line 134-136: What are the categories?

Tables:

-Wealth index: what are the ranges of poor, moderate ….by numbers?

- Footnotes should be added to your tables

-

Discussion:

Line 202: Add few studies as you have mentioned in the text.

Line 208, 209: rephrase the sentence using DMFT term.

Line 223: Add reference

Line 231: Reference 33, 34 are irrelevant

Line 242: Discuss further regarding developing country and provide reference.

Line 253: Few studies… (however one reference was mentioned)

Line 258: Add reference

Line 258-260: The discussion is weak regarding the tobacco consumption

Discussion section needs improvement and more details and justifications for your findings should be presentd.

General comments:

English language needs to be checked.

It is nice to see at least one figure.

References need to be checked as many sentences need referencing, and some references are not really relevant to the text.

6. PLOS authors have the option to publish the peer review history of their article (what does this mean?). If published, this will include your full peer review and any attached files.

Reviewer #1: No

Reviewer #2: No

---

## [Author Response · Author response to Decision Letter 0]

16 Jun 2023

Response to Reviewers:

Reviewer 1:

Introduction

• A major drawback of this manuscript is that the authors failed to review literature on GERD and dental caries. In addition, they didn’t identify any gap/s in literature that their study would want to address.

Response: The text was edited as suggested by the reviewer.

Page 3, Lines 64¬¬¬¬–70; Page 5, Lines 92¬¬¬¬–100.

• Generally, the paragraphs don’t flow into each other – the previous paragraph doesn’t flow to the next.

Response: The text editing has been performed as suggested by the reviewer.

Methods

• In addition to other confounding variables, the authors should have considered the oral hygiene status of the participants. 

• Similarly, there is no information on the fluoride exposure of the participants in the study.

Response: The present study is a subsample of the data of an epidemiological study with a sample size of 15,000 people (Azar Cohort Study). Due to the large sample size, some variables, such as oral hygiene status and fluoride intake, were not investigated. 

Also, due to the inclusion criteria of the Azar Cohort Study— permanent residence in the Shabestar district—it can be concluded that they were exposed to fluoride in the same conditions.

• Although information on free sugar consumption was obtained, the frequency of free sugar consumption between meals will be more appropriate in relation to caries formation.

Response: Based on studies by Sheiham (2001) and Moynihan (2014), both the frequency of consumption and the total amount of free sugars are important in the etiology of caries. Considering the common risk factor approach, in the present study, the amount of free sugars intake was used instead of the frequency of consumption.

A. Sheiham. Dietary effects on dental diseases. Public Health Nutrition. 2001.4(2B),569-591.

P.J. Moynihan and S.A.M. Kelly, Effect on Caries of Restricting Sugars Intake: Systematic Review to Inform WHO Guidelines. Journal of Dental Research. 2014. 93(1):8-18.

• Classification of GERD frequency of regurgitation should be referenced.

Response: It was added on page 8, line 164.

According to the classification of Wenzl 2020 (Never, Less than weekly, Weekly, Daily) and considering the data distribution in the sample, it was classified into three groups. 

E.M. Wenzl, R. Ried, A. Borenich, W. Petritsch, H.H. Wenzl. Low prevalence of gastroesophageal reflux symptoms in vegetarians. Indian Journal of Gastroenterology. 

GERD comprises a broad spectrum of disorders. The typical symptoms of GERD in adult patients are retrosternal or sub-sternal burning, regurgitation, epigastric pain, and dysphagia. Symptoms related to the reflux of gastric contents into the esophagus – principally heartburn and regurgitation – are widespread in the general population. Regurgitation, the perception of gastric contents moving from the stomach into the esophagus, experienced by 40–50% of patients with reflux disease, is also a typical symptom. 

R. JONES & J. P. GALMICHE. Review: What do we mean by GERD? Definition and diagnosis. Alimentary pharmacology & therapeutics. 2005; 22 (Suppl. 1): 2–10.

Results

• It would have been better to compare the mean DMFT scores of both groups based on the different variables (socio-demographic, oral health practices and BMI) before the regression analysis.

Response: Table 2 was added according to the reviewer's suggestion.

Page 12, Line 235.

Discussion

• This is inadequate- the authors didn’t discuss the reasons behind their findings.

Response: The discussion section was expanded, and more details and justifications were discussed.

Page 15, Lines 267–272; Page 15, Lines 281–283; Pages 15 & 16, Lines 291–294; Page 16, Lines 311–313.

Reviewer 2:

Title

• It is limited, as there are many other factors that have been checked other than GERD. So you need to modify your title accordingly.

Response: According to the primary objective of this study, the main independent variable that might affect dental health was GERD. The variables potentially affecting this relationship, considering confounding effects, include demographic factors (age and gender), socioeconomic status (wealth score index and educational level), behavioral habits (tobacco use, alcohol consumption, frequency of toothbrushing, and consumption of sweet foods and beverages containing free sugars), and BMI.

Abstract

• Objectives: There are many other objectives that were not mentioned.

Response: According to the primary objective of this study, the main independent variable that might affect dental health was GERD. The other variables potentially affecting this relationship, considering confounding effects, include: demographic factors (age and gender), socioeconomic status (wealth score index and educational level), behavioral habits (tobacco use, alcohol consumption, frequency of toothbrushing, and consumption of sweet foods and beverages containing free sugars), and BMI.

• Materials and methods: very confusing, as I don’t understand if all your data obtained from dataset previously recorded, Or the clinical examination and the interviews were conducted as part of this study. This same comment is applied to methodology section that needs to be re-written.

Response: The “Methods” section was rewritten for more clarity. All data used in the present study was obtained from the dataset of ACS and had been recorded previously. 

Introduction 

• Line 53: move word ‘still’ after ‘are’.

Response: The text was edited as suggested by the reviewer.

Page 3, Line 55.

• Line 60: consistency in writing word ‘socioeconomic’.

Response: The text was edited as suggested by the reviewer.

Page 3, Line 62.

• Next to first paragraph, additional paragraph should be added to briefly preset the socioeconomic and environmental factors that is related or may affect dental caries.

Response: The text was edited as suggested by the reviewer.

Page 3 & 4, Lines 62–70.

• Line 78: Add reference.

Response: It was added.

Page 4, Line 88.

• Add null hypothesis to the last paragraph.

Response: It was added.

 Page 5, lines 103–104. 

Methods

• Very confusing as mentioned previously, which data that have been obtained from previous records and which data were obtained by clinical examination and questionnaires in this study? (This same comment is applied to methodology section that needs to be re-written).

Response: The “Methods” section was rewritten for more clarity. All the data used in this analytical cross-sectional study were obtained from the enrollment phase of the ACS and had been recorded previously.

• And if questionnaires were used, you have to specify is it online, face to face interview, paper based?

Response: It was addressed on page 6, Line 115.

Four questionnaires (one general, two medical, and one nutrition) consisting of 482 items were used to collect information in the Azar Cohort Study; and face-to-face interviews and clinical examinations were used to complete these questionnaires. Data are recorded online and stored in a centralized database and in 3 other locations, being backed up every 30 minutes. 

• What was the response rate in each questionnaire, number of items in each questionnaire, how these items were validated, any piloting for the questionnaire was done?

Response: Since face-to-face interviews and clinical examinations were used to complete these questionnaires and the information was registered online in the cohort study databank, the response rate was 100%. All questionnaires are checked for completeness by field supervisors. It was addressed on page 6, Lines 115–118.

The Azar Cohort Study was set up in three phases: (i) pilot study; (ii) enrolment of participants; and (iii) regular follow-up of subjects for 15 years (Farhang 2019). The aim of the pilot phase was to appraise the feasibility of the study and reveal any unmet needs for undertaking the full-scale AZAR cohort study. Specifically, the pilot phase aimed to implement valid and reproducible methods and test the structured questionnaires. Also, the items in the questionnaires were categorized and presented in their own subgroups and were not numbered.

Poustchi H, Eghtesad S, Kamangar F, Etemadi A, Keshtkar AA, Hekmatdoost A, et al. Prospective Epidemiological Research Studies in Iran (the PERSIAN Cohort Study): Rationale, Objectives, and Design. Am J Epidemiol. 2018;187(4):647-655.

Farhang S, Faramarzi E, Amini Sani N, Poustchi H, Ostadrahimi A, Alizadeh BZ, Somi MH. Cohort Profile: The AZAR cohort, a health-oriented research model in areas of major environmental change in Central Asia. Int J Epidemiol. 2019; 48(2):382-382h.

• Why 905 were selected from 2010 patients with GERD, and how these patients were selected.

Response: According to the exclusion criteria, subjects with full dentures were excluded from the present study.

It was addressed on page 6, Line 133.

A total of 2010 patients with GERD were identified in the Azar cohort population, and 905 subjects without full dentures qualified for the case group. Fig 1 presents the selection process of the study subjects.

• GERD patients only were diagnosed based on the two questions that have been mentioned.

Response: The present study is a subsample of the data of a large epidemiological study with a sample size of 15,000 people (Azar Cohort Study). Due to the large sample size, data were collected by questionnaires. 

GERD comprises a broad spectrum of disorders. The typical symptoms of GERD in adult patients are retrosternal or sub-sternal burning, regurgitation, epigastric pain, and dysphagia. Regurgitation, the perception of gastric contents moving from the stomach into the esophagus, experienced by 40–50% of patients with reflux disease, is also a typical symptom (Jones 2005). 

R. JONES & J. P. GALMICHE. Review: What do we mean by GERD? Definition and diagnosis. Alimentary pharmacology & therapeutics. 2005; 22 (Suppl. 1): 2–10.

• What do you mean by oral examiner? This is not acceptable description.

Response: The text was edited as suggested by the reviewer.

Page 7, Line 152.

• Variables and data measurement section is not clear and more details should be added.

Response: The “Variables and data measurement” section was rewritten, and some details were added for more clarity.

Page 7 & 8, Lines 149–177.

• Line 134-136: What are the categories?

Response: It was addressed in the text.

Page 8, Lines 174–175.

Tables

• Wealth index: what are the ranges of poor, moderate ….by numbers?

Response: It has been addressed in the text.

Page 8, Lines 169–171.

• Footnotes should be added to your tables

Response: The tables were revised as suggested by the reviewer.

Pages 10, 12 & 13.

Discussion

• Line 202: Add few studies as you have mentioned in the text.

Response: It was added.

 Page 14, Line 245.

• Line 208, 209: rephrase the sentence using DMFT term.

Response: The text was edited as suggested by the reviewer.

Page 14, Line 252.

• Line 223: Add reference

Response: It was added.

Page 15, Line 267.

• Line 231: Reference 33, 34 are irrelevant.

Response: The references were revised as suggested by the reviewer.

Page 15, Line 280.

• Line 242: Discuss further regarding developing country and provide reference.

Response: It was addressed on page 15 & 16, Lines 291–294.

• Line 253: Few studies… (However one reference was mentioned)

Response: The reference provided for this section is a systematic review study to Inform WHO Guidelines and was published in 2014. It is mentioned in this article that “The majority of studies identified were conducted on children, while only four studies were on adults".

Moynihan PJ, Kelly SA. Effect on caries of restricting sugars intake: systematic review to inform WHO guidelines. J Dent Res. 2014;93(1):8-18.

• Line 258: Add reference

Response: It was added on page 16, line 311.

According to the Fejerskov and Manji model for caries causation, at the individual level and in the presence of adequate saliva, optimum fluoride exposure and nutritional pattern are important determining factors for the development of caries.

• Line 258-260: The discussion is weak regarding the tobacco consumption. 

Response: The text was revised as suggested by the reviewer.

Page 16, Lines 311–312

• Discussion section needs improvement and more details and justifications for your findings should be presented.

Response: The discussion section was expanded, and the reasons behind the findings were discussed.

General comments

• English language needs to be checked.

Response: The English text was edited professionally.

• It is nice to see at least one figure.

Response: It was added on page 7, Line 140.

• References need to be checked as many sentences need referencing, and some references are not really relevant to the text.

Response: References were revised and irrelevant references were deleted.

---

## [Decision Letter · Decision Letter 1]

10 Jul 2023

PONE-D-23-10589R1Association between gastroesophageal reflux disease and dental caries among adults in the Azar cohort population: A cross-sectional studyPLOS ONE

Dear Dr. Mahboobi,

Thank you for submitting your manuscript to PLOS ONE. After careful consideration, we feel that it has merit but does not fully meet PLOS ONE’s publication criteria as it currently stands. Therefore, we invite you to submit a revised version of the manuscript that addresses the points raised during the review process.

We look forward to receiving your revised manuscript.

Kind regards,

Hadi Ghasemi

Academic Editor

PLOS ONE

Journal Requirements:

Reviewers' comments:

Reviewer's Responses to Questions

**Comments to the Author**

1. If the authors have adequately addressed your comments raised in a previous round of review and you feel that this manuscript is now acceptable for publication, you may indicate that here to bypass the “Comments to the Author” section, enter your conflict of interest statement in the “Confidential to Editor” section, and submit your "Accept" recommendation.

Reviewer #2: (No Response)

Reviewer #3: All comments have been addressed

Reviewer #4: (No Response)

2. Is the manuscript technically sound, and do the data support the conclusions?

Reviewer #2: Yes

Reviewer #3: Yes

Reviewer #4: Yes

3. Has the statistical analysis been performed appropriately and rigorously? 

Reviewer #2: Yes

Reviewer #3: Yes

Reviewer #4: Yes

4. Have the authors made all data underlying the findings in their manuscript fully available?

Reviewer #2: No

Reviewer #3: Yes

Reviewer #4: No

5. Is the manuscript presented in an intelligible fashion and written in standard English?

Reviewer #2: Yes

Reviewer #3: Yes

Reviewer #4: Yes

6. Review Comments to the Author

Reviewer #2: Thank you for addressing all comments, few comments were added to the revised manuscript itself. Thank you

Reviewer #3: All comments by previous reviewer have been addressed adequately. The methods have been addressed comprehensively, the discussion has been improved upon. Also the English has been adequately corrected. This new reviewed article makes a better read.

Reviewer #4: General Comments:

The paper "Association between gastroesophageal reflux disease and dental caries among adults in the Azar cohort population: A cross-sectional study" presents a study that looked at the relationship between GERD and dental caries using the DMFT index. The study investigates potential confounding variables and gives a thorough analysis of the findings. Overall, the work is well-written and structured, and it contributes significantly to the subject of dentistry. While some small edits and clarifications are needed, the manuscript gives a compelling research study with relevant findings.

Specific Comments:

Introduction:

The introduction gives a clear context and rationale for the research. Given that tooth erosion is a prevalent issue in GERD patients, it effectively emphasizes the necessity of knowing the relationship between GERD and dental caries. The introduction also discusses the existing literature gap about the relationship between GERD and dental caries. However, at the end of the introductory section, a more clear and focused statement of the study's aims would be beneficial with clear hypothesis.

Methods:

The methods section discusses the study's design, participant selection, and data collection procedures in detail. Given its widespread usage in dentistry research, the use of the DMFT index as a measure of dental caries is justified. However, some small clarifications are required. For example, more information on how GERD was assessed, such as the questions or criteria utilized in the questionnaire, would be beneficial. The limitations of self-reported data, as well as their potential impact on the study's conclusions, should also be addressed.

Results:

The results section is well-organized and presents the study's findings clearly. The statistical analysis, including regression analysis, strengthens the study. However, for significant findings, it would be beneficial to provide the effect sizes to provide a better understanding of the magnitude of the associations observed.

Discussion:

The discussion section explores the study's findings in depth in relation to previous studies, noting both consistent and contradictory findings. The discussion is enriched by the inclusion of numerous aspects that may contribute to the association between GERD and dental caries, such as salivary flow, swallowing difficulty, and oral hygiene behaviors. However, a more extensive examination of the potential processes behind the observed relationships, particularly those related to the decline in cariogenic bacteria despite lower saliva quality and quantity, would be beneficial. Furthermore, the study's limitations, such as its cross-sectional design and dependence on self-reporting, are noted correctly. The conclusion emphasizes the need for additional research, which is appropriate.

Language and Structure:

The manuscript is generally well-written with clear and concise language.

Figures and Tables:

The figures and tables provided in the manuscript are relevant and contribute to the understanding of the results.

Overall, the manuscript makes an important contribution to the field of dentistry by exploring the relationship between GERD and dental caries. The study is well-designed, and the results are effectively presented and analyzed.

7. PLOS authors have the option to publish the peer review history of their article (what does this mean?). If published, this will include your full peer review and any attached files.

Reviewer #2: No

Reviewer #3: **Yes: **ADEBAYO PETER Adewuyi

Reviewer #4: **Yes: **Furqan Ahmed

---

## [Author Response · Author response to Decision Letter 1]

24 Jul 2023

Response to Reviewers:

Reviewer 2:

1. What do you mean by proximal and distal determinants?

Response: “The chain of events leading to an adverse health outcome can be both proximal and distal; proximal factors act directly or almost directly to cause diseases, while distal factors are further back in the causal chain and act via a number of intermediary causes.”

Petersen PE. Sociobehavioural risk factors in dental caries – international perspectives. Community Dent Oral Epidemiol 2005; 33: 274–9.

2. Add reference (page 4, line 65).

Response: The content of this paragraph is from chapter 4 of the book "Dental Caries: the disease and its clinical management" which is about the epidemiology of dental caries. The book has been referenced at the end of the paragraph.

3. Re-phrase, it is not clear. 

Response: It was re-phrased (page 4, line 67) 

These conditions provide the opportunities for and limit individual behaviors.

4. What are these numbers, which currency? (Line 169)

Response: It’s not a currency. The Wealth Score Index (WSI) in the Azar Cohort Study was based on assets such as dishwashers, vehicles, and flat-screen TVs as well as house conditions (e.g. the number of rooms, and type of ownership). This index was determined via Multiple Correspondence Analysis (MCA) and grouped into five quintiles, the first being the lowest and the fifth being the highest score.

Reviewer 4:

1. Introduction:

The introduction gives a clear context and rationale for the research. Given that tooth erosion is a prevalent issue in GERD patients, it effectively emphasizes the necessity of knowing the relationship between GERD and dental caries. The introduction also discusses the existing literature gap about the relationship between GERD and dental caries. However, at the end of the introductory section, a more clear and focused statement of the study's aims would be beneficial with clear hypothesis.

Response: It was addressed on page 5, line 101:

This study hypothesizes that there is no association between having GERD and dental caries development considering potential confounders.

2. Methods:

The methods section discusses the study's design, participant selection, and data collection procedures in detail. Given its widespread usage in dentistry research, the use of the DMFT index as a measure of dental caries is justified. However, some small clarifications are required. For example, more information on how GERD was assessed, such as the questions or criteria utilized in the questionnaire, would be beneficial. The limitations of self-reported data, as well as their potential impact on the study's conclusions, should also be addressed.

Response: It was done; page 7, lines 148-150.

- Among the variables of the Azar cohort study used in the present study, dental caries and body mass index (BMI) were measured based on clinical examinations by trained examiners. Dental caries was recorded as the number of decayed, missing, and filled teeth (DMFT index) according to WHO criteria [27], and the examiner was uninformed of the systemic conditions of the participants. 

Other variables of the present study, including having GERD, were determined using a questionnaire and based on the participants' self-report (It has been addressed on pages 7 & 8; lines 152-161). The criterion for having GERD was the answer to these two questions: “Have you ever been diagnosed with gastroesophageal reflux disease?” and “Have you had reflux of food from the stomach to the esophagus in the past year?” (It has been addressed on page 6; lines 126-129).

The limitations of the study design and self-reported data have been considered in Discussion section (page 17, lines 324-329). 

3. Results:

The results section is well-organized and presents the study's findings clearly. The statistical analysis, including regression analysis, strengthens the study. However, for significant findings, it would be beneficial to provide the effect sizes to provide a better understanding of the magnitude of the associations observed.

Response: It was addressed on page 11; line 223.

4. Discussion:

The discussion section explores the study's findings in depth in relation to previous studies, noting both consistent and contradictory findings. The discussion is enriched by the inclusion of numerous aspects that may contribute to the association between GERD and dental caries, such as salivary flow, swallowing difficulty, and oral hygiene behaviors. However, a more extensive examination of the potential processes behind the observed relationships, particularly those related to the decline in cariogenic bacteria despite lower saliva quality and quantity, would be beneficial. Furthermore, the study's limitations, such as its cross-sectional design and dependence on self-reporting, are noted correctly. The conclusion emphasizes the need for additional research, which is appropriate.

Response: It was done; page 15, lines 265-267.

Filipi explained that although S. mutans can survive in pH values <4.2, the pH decline in the oral cavity of GERD might be so remarkable that it can stop the metabolic activity of S. mutans.

With best regards and thanks for the careful review and valuable comments of the reviewers.

---

## [Editor Report · Decision Letter 2]

26 Jul 2023

Association between gastroesophageal reflux disease and dental caries among adults in the Azar cohort population: A cross-sectional study

PONE-D-23-10589R2

Dear Dr. Zeinab Mahboobi,

We’re pleased to inform you that your manuscript has been judged scientifically suitable for publication and will be formally accepted for publication once it meets all outstanding technical requirements.

Kind regards,

Hadi Ghasemi

Academic Editor

PLOS ONE
---

## [Editor Report · Acceptance letter]

31 Jul 2023

PONE-D-23-10589R2 

*Association between gastroesophageal reflux disease and dental caries among adults in the Azar cohort population: A cross-sectional study*

Dear Dr. Mahboobi:

I'm pleased to inform you that your manuscript has been deemed suitable for publication in PLOS ONE. Congratulations! Your manuscript is now with our production department. 

Kind regards, 

on behalf of

Dr. Hadi Ghasemi 

Academic Editor

PLOS ONE